# Epistatic interactions between killer immunoglobulin-like receptors and human leukocyte antigen ligands are associated with ankylosing spondylitis

**Aimee L. Hanson**[1], **International Genetics of Ankylosing Spondylitis Consortium**[¶], **Damjan Vukcevic**[2,3], **Stephen Leslie**[2,3,4], **Jessica Harris**[5,6], **Kim-Anh Lê Cao**[2], **Tony J. Kenna**[5,6], **Matthew A. Brown**[5,6¤]*

**1** University of Queensland Diamantina Institute, University of Queensland, Brisbane, Queensland, Australia, **2** Melbourne Integrative Genomics, School of Mathematics and Statistics, University of Melbourne, Parkville, Victoria, Australia, **3** Data Science, Murdoch Children's Research Institute, Parkville, Victoria, Australia, **4** School of Biosciences, University of Melbourne, Parkville, Victoria Australia, **5** Institute of Health and Biomedical Innovation, Queensland University of Technology, Brisbane, Queensland, Australia, **6** Translational Research Institute, Princess Alexandra Hospital, Brisbane, Queensland, Australia

¤ Current address: Guy's and St Thomas' NHS Foundation Trust and King's College London NIHR Biomedical Research Centre, King's College London, London, England.
¶ Members of The International Genetics of Ankylosing Spondylitis Consortium are provided in the Acknowledgments.
* matt.brown@kcl.ac.uk

## Abstract

The killer immunoglobulin-like receptors (KIRs), found predominantly on the surface of natural killer (NK) cells and some T-cells, are a collection of highly polymorphic activating and inhibitory receptors with variable specificity for class I human leukocyte antigen (HLA) ligands. Fifteen *KIR* genes are inherited in haplotypes of diverse gene content across the human population, and the repertoire of independently inherited *KIR* and *HLA* alleles is known to alter risk for immune-mediated and infectious disease by shifting the threshold of lymphocyte activation. We have conducted the largest disease-association study of *KIR*-*HLA* epistasis to date, enabled by the imputation of *KIR* gene and *HLA* allele dosages from genotype data for 12,214 healthy controls and 8,107 individuals with the *HLA-B\*27*-associated immune-mediated arthritis, ankylosing spondylitis (AS). We identified epistatic interactions between *KIR* genes and their ligands (at both *HLA* subtype and allele resolution) that increase risk of disease, replicating analyses in a semi-independent cohort of 3,497 cases and 14,844 controls. We further confirmed that the strong AS-association with a pathogenic variant in the endoplasmic reticulum aminopeptidase gene *ERAP1*, known to alter the HLA-B\*27 presented peptidome, is not modified by carriage of the canonical HLA-B receptor KIR3DL1/S1. Overall, our data suggests that AS risk is modified by the complement of *KIR*s and *HLA* ligands inherited, beyond the influence of *HLA-B\*27* alone, which collectively alter the proinflammatory capacity of KIR-expressing lymphocytes to contribute to disease immunopathogenesis.

**Data Availability Statement:** All numerical data are within the manuscript and supporting information files.

**Funding:** MB is supported by an Australian National Health and Medical Research Council (NHMRC; https://www.nhmrc.gov.au/) Senior Principal Research Fellowship (grant #1024879). AH was supported by an Australian Government Research Training Program Stipend (https://www.education.gov.au/). SL is supported by a NHMRC Career Development Fellowship (grant #1053756) and a Victorian Life Sciences Computation Initiative (https://www.melbournebioinformatics.org.au/; grant #VR0240) funding the Peak Computing Facility at the University of Melbourne. Research at the Murdoch Children's Research Institute was supported by the Victorian Government's Operational Infrastructure Support Program (https://www2.health.vic.gov.au/). K-ALC was supported in part by a NHMRC Career Development Fellowship (grant #1159458). The funders had no role in study design, data collection and analysis, decision to publish, or preparation of the manuscript.

**Competing interests:** The authors have declared that no competing interests exist.

## Author summary

Cells of the immune system utilise various cell-surface receptors to differentiate between healthy and infected or malignant cells, enabling targeted inflammatory responses while minimising damage to self-tissue. In instances where the immune system fails to correctly differentiate healthy from diseased tissue, or inflammatory activity is poorly regulated, autoimmune or autoinflammatory conditions can develop. Here we have investigated a possible role for a class of immune-cell activating and inhibitory receptors in the pathogenesis of ankylosing spondylitis (AS), a common but poorly understood inflammatory arthritis in which the immune system causes severe damage to the joints of the pelvis and spine. Using genetic information from 12,214 healthy controls and 8,107 individuals with AS we were able to identify combinations of independently inherited immune cell receptors and their ligands that increase or decrease an individual's risk of disease. This research provides new insight into the nature of co-inherited genetic factors that may collectively alter the proinflammatory capacity of immune cells, contributing to the immunopathogenesis of immune-mediated diseases.

## Introduction

Ankylosing spondylitis (AS) is an immune-mediated arthritis in which inflammation targeting particularly the pelvis and spine contributes to joint erosion and reactive bone deposition. The disease is strongly associated with inheritance of the human leukocyte antigen (HLA) class I allele *HLA-B*27* (carried by >80% of patients) [1, 2], as well as polymorphisms in the endoplasmic reticulum aminopeptidases (ERAP1 and ERAP2) [3–5] involved in trimming endogenous peptides for HLA class I presentation. Despite a thoroughly characterised genetic architecture, implicating >100 loci in disease pathogenesis [3–9], the immunological mechanisms underlying AS are not fully understood. Of interest is the potential role of killer immunoglobulin-like receptors (KIRs) in disease, a diverse collection of paired signalling receptors expressed predominantly on natural killer (NK) cells (and some T-cell subpopulations) that exhibit variable specificity for class I HLA ligands [10–12]. Opposing inhibitory and activating signals transduced through surface KIRs buffer the threshold of activation of KIR-expressing cells, particularly NK cells, serving to quench or promote innate killing activity as required to maintain immune homeostasis and control inflammatory responses. The hypervariable nature of KIRs and their HLA ligands bestows upon the human population a large spectrum of proficiencies in lymphocyte responses to activating stimuli and self-tolerance. Accordingly, KIR-HLA co-inheritance has been associated with various immune-related phenotypes, including infection outcome [13–15], and susceptibility to autoimmunity [16–18] and cancer [19, 20].

Fifteen *KIR* genes have been identified in humans, all of which share significant sequence homology (85–99% similarity) attributed to the duplication and progressive evolution of a common ancestral gene [21, 22]. The *KIR* locus (chromosome 19q13.4) exhibits extreme copy number variation, with genes inherited in haplotypes of diverse content that often carry duplication, deletion and hybridisation events [22, 23]. Classically, *KIR* haplotypes have been categorised into two groups; group A haplotypes containing the single activating receptor gene *KIR2DS4* among six inhibitory receptors, and B haplotypes carrying a variable number of both activating and inhibitory receptor genes [24]. Stochastic expression of inherited KIRs is largely controlled by variable promoter methylation, with the vertical transmission of methylation

patterns enabling clonal populations of NK cells to maintain established KIR expression patterns over cell divisions [25, 26]. Immense allelic polymorphism further magnifies interindividual *KIR* diversity, with functional consequences for receptor expression, signalling strength and ligand affinity [27–31]. Considering both allelic and gene content diversity at the *KIR* locus, it is improbable that any two unrelated individuals carry an identical complement of *KIR* alleles.

Inhibitory KIRs recognise constitutively expressed HLA class I ligands on host cells. Under homeostatic conditions, persistent suppressive signalling transduced through phosphorylation of immunoreceptor tyrosine-based inhibitory (ITIM) motifs within the receptor cytoplasmic domain quench innate NK cytotoxic activity, safeguarding against unchecked autoimmunity [32]. Conversely, HLA downregulation on target cells (during viral infection or transformation of tumour cells) facilitates NK killing in the absence of co-stimulation [33]. Despite a conserved mode of ligand recognition [12], different inhibitory KIRs engage HLA class I subtypes with varying affinities and exclusions, and exhibit a degree of specificity for the bound peptide [34–37]. KIR3DL1 recognises approximately one third of HLA-B and 20% of HLA-A molecules carrying the Bw4 motif (defined by amino-acids at positions 77–83 in the α1 domain) [38, 39], the strongest inhibition conferred by HLA-Bw4 ligands with a position 80 isoleucine (Bw4I80) [40]. KIR3DL2 recognises HLA-A*03 and A*11 [36]. KIR2DL2 and KIR2DL3 recognise HLA-C molecules of the HLA-C1 subclass expressing asparagine at position 80 (Asn80), and KIR2DL1 those of the HLA-C2 subclass (Lys80), though some KIR2LD2 cross reactivity with HLA-C2 has been reported [41]. Conversely, the ligands for activating KIRs are poorly defined, with these receptors exhibiting a far weaker affinity for HLA than their inhibitory homologues [37, 42, 43]. Evidence for engagement of ligands upregulated on cancer cell lines independent of HLA expression [44], non-classical HLA molecules expressed by proliferating lymphoid and monocytic cells under inflammatory conditions [45, 46], and HLA-restricted viral epitopes [47], suggests that activating KIR signalling may be most productive in instances of cellular stress and infection. Known HLA ligands for activating and inhibitory KIRs, and HLA subclass groupings, are listed in **Table 1**.

Studies across ethnic groups have reported differential *KIR* inheritance in AS cohorts [57–66], for the most part consolidating disease-associations with *KIR3DL1*, encoding the only known inhibitory receptor for HLA-Bw4 subclass ligands (including AS-risk subtypes *HLA-B*27:01,:02,:03,:04,:05* and: *07,*), and its activating homologue and alternate allele *KIR3DS1*. Underrepresentation of *KIR3DL1* has been reported in *HLA-B*27+ patients relative to *HLA-B*27+ controls [57, 58, 61, 64], whereas carriage of *KIR3DS1* and *KIR3DL1|KIR3DS1* heterozygosity has been associated with increased disease risk [57–59, 61, 64], implying a dominant role for *KIR3DS1* as a genetic risk factor. However, these studies have all had modest sample sizes, the strongest associations have been suggestive only, and no study reported to date has controlled for population stratification. These findings have also not been uniformly replicated, with a large study of ~600 AS patients and controls from the United Kingdom reporting no *KIR* associations with disease [60], and others reporting varying *KIR-HLA-C* co-associations [63, 65, 66]. Disparate results may be attributed to inconsistencies in study conduct, sample size or ethnic background. Furthermore, *KIR3DL1* is among the most polymorphic *KIR*s [67], encoding allotypes with drastically varied expression profiles (null, low and high) [68] and strength of ligand binding [27, 29, 56] that have largely been overlooked by genetic studies of *KIR*-associations with AS to date.

Beyond association studies, there is presently no functional evidence that the primary role of canonical HLA-B*27 in AS can be ascribed to molecular interactions with activating or inhibitory KIRs. However, non-canonical HLA-B*27 homodimers (B27$_2$) and monomeric free HLA-B*27 heavy chains (FHCs) have been found to interact with KIR3DL1 and KIR3DL2 [69,

**Table 1. Known ligands for activating and inhibitory KIRs and subclass classification of HLA alleles.**

| KIR | | HLA Ligand | Ref |
|---|---|---|---|
| **INHIBITORY** | KIR2DL1 | **HLA-C2 (Lys80)** | [42] |
| | KIR2DL2/3 | **HLA-C1 (Asn80)**, HLA-B*46:01, HLA-B*73:01 and some HLA-C2 (low affinity) | [43, 48] |
| | KIR2DL4[a] | **HLA-G** | [49] |
| | KIR2DL5 | Unknown | |
| | KIR3DL1 | **HLA-Bw4B** (I80 > T80) including some **HLA-Bw4A** (A*23, A*24, A*25 and A*32) | [40] |
| | KIR3DL2 | **HLA-A*03, HLA-A*11** and HLA-B*27 homodimers | [50–52] |
| | KIR3DL3 | Unknown | |
| **ACTIVATING** | KIR2DS1 | **HLA-C2 (Lys80)** | [42] |
| | KIR2DS2 | **HLA-A*11, HLA-C1 (Asn80)** (low affinity) | [53, 54] |
| | KIR2DS3 | Unknown | |
| | KIR2DS4 | **HLA-A*11** and some HLA-C (C*01, C*02, C*04, C*05, C*14, C*16) | [55] |
| | KIR2DS5 | Unknown | |
| | KIR3DS1 | Possibly **HLA-Bw4** (low affinity, peptide dependent) | [47, 56] |
| **HLA subclass** | | **Alleles** | |
| HLA-C1 | | HLA-C*01, C*03, C*07, C*08, C*12, C*13, C*14, C*16 and HLA-B*46:01, B*73:01 (Asparagine at position 80 in the alpha-helix; HLA-C Asn80.) | |
| HLA-C2 | | HLA-C*02, C*04, C*05, C*06, C*15, C*17, C*16:02, C*18 (Lysine at position 80 in the alpha-helix; HLA-C Lys80.) | |
| HLA-Bw4I80 | | HLA-B*15:13, 15:16, B*15:17, B*15:23, B*15:24, B*27:02, B*38:01, B*49, B*51, B*52, B*53, B*57, B*58, B*59, HLA-A*23, A*24, A*25, A*32 (Bw4 is defined by an epitope at position 77–83 in the alpha1-helix) | |
| HLA-Bw4T80 | | HLA-B*13, B*27:01, B*27:03, B*27:04, B*27:05, B*27:07, B*37, B*38:02, B*44, B*47 (Bw4 is defined by an epitope at position 77–83 in the alpha1-helix) | |
| HLA-Bw6 | | HLA-B*07, B*08, B*14, B*15:01, B*15:02, B*15:03, B*15:05, B*15:08, B*15:09, B*15:10, B*15:14, B*15:15, B*15:18, B*15:25, B*15:29, B*18, B*22, B*27:08, B*35, B*39, B*40, B*41, B*42, B*45, B*46, B*48, B*50, B*54, B*55, B*56, B*60, B*61, B*62, B*64, B*65, B*67, B*70, B*71, B*72, B*73, B*75, B*76, B*78, B*81, B*82. | |
| N.B. Some HLA alleles differ in their Bw4I80/T80/Bw6 classification at four-digit resolution. | | | |

[a] KIR2DL4 is expressed on endosomes and despite having a long cytoplasmic tail containing an ITIM inhibitory signalling motif it transduces activating signals upon coupling with accessory protein FcRγ.

70], with the KIR3DL2 receptor upregulated on CD4+ T-cell populations in spondyloarthritis patients and shown to prompt differentiation in to pathogenic Th17 cells upon B27$_2$ ligation [71]. Intriguingly, loss of HLA-B*27:05 recognition by KIR3DL1+ NK cells drives target cell killing in peptide-specific contexts [34], posing the hypothesis that features of a disease-specific peptidome may disrupt canonical inhibitory KIR interactions. In support of this, disease-associated polymorphisms in *ERAP1* alter class I peptide production and destruction and exhibit strong genetic epistasis with *HLA-B*27 [4]. Alternatively, it is plausible that the complement of coinherited *KIR* and *HLA* alleles collectively modifies AS risk by shifting the activation threshold of lymphocyte populations, exacerbating autoinflammation.

Understanding of the genetic contribution of KIRs to variability in immunological phenotypes has lagged substantially behind the major histocompatibility complex. Statistical methods to accurately impute *HLA* alleles from single nucleotide polymorphisms (SNPs) have revolutionised the accessibility of this locus for large scale disease-association studies. More recently, the development of the *KIR* imputation technique, KIR*IMP, has enabled phased *KIR* gene content haplotypes to be inferred from genotyping data alone, providing a means to assess variable *KIR* inheritance in powerful study cohorts [72]. Here, we have imputed *KIR* haplotypes from Immunochip genotype data available for 8,107 AS cases and 12,214 healthy controls from the International Genetics of Ankylosing Spondylitis (IGAS) consortium [5],

enabling the largest *KIR* disease-association study conducted to date. Pairing *HLA* allele and *KIR* dosage information we have identified disease-associated epistatic interactions between *KIR* genes and their ligands (at both HLA subtype and allele resolution), replicating analyses in a semi-independent cohort of 3,497 cases and 14,844 controls, and tested for evidence of a three-way interaction between *KIR3DL1/S1*, *ERAP1* and *HLA-B*$^*$*27* in disease. Additionally, we have employed a PCR approach [73] in a subset of samples to distinguish six function groups of KIR3DL1 allotypes that differ in surface expression and strength of HLA-Bw4 binding, addressing a possible role for KIR3DL1 variants in HLA-B*27+ AS.

## Materials and methods

### Ethics and sample acquisition

The IGAS test cohort comprised 8,107 AS cases and 12,214 healthy controls of Caucasian decent, originally recruited by the IGAS Consortium as reported in Cortes *et al.* 2013 [5]. Written informed consent was obtained from all participants with research ethics approval granted by the relevant ethics committee at each participating centre [5]. All cases had an AS diagnosis as defined by the modified New York criteria [74]. A semi-independent replication cohort comprised 14,844 controls sourced from the UK Biobank (UKB) public resource (project 21024), and the subset of 3,497 IGAS AS cases recruited from the UK, also included in the test cohort. UKB controls, aged between 40–69 years, were selected through exclusion of individuals coding as having one of the following disease states: ankylosing spondylitis (code 1313), inflammatory bowel disease (code 1461), Crohn's disease (code 1462), ulcerative colitis (code 1463), psoriasis (code 1453) or spine arthritis/spondylitis (code 1311); a kinship coefficient $> = 0.0442$ and non-Caucasian ancestry, with 20,000 of the remaining identifiers selected by random number generation for study inclusion prior to quality control filtering as described below and in **Fig 1**.

### Genotyping

IGAS test cohort samples were genotyped using the Illumina Immunochip array on the Illumina Infinium platform as previously described [5]. UKB replication cohort controls were genotyped on the UKB Affymetrix Axiom array. Identity by decent was calculated using the— `genome` command in PLINK [75], with exclusion of one individual from each pair with a PI_HAT score $>0.05$. Principal components (PCs) for ethnicity confirmation and population stratification correction were calculated for test and replication cohorts independently based on 20,783 and 12,485 autosomal SNPs respectively outside of long-range LD regions [76]. Homogeneity of ethnic background was confirmed by visualisation of the first two PCs with exclusion of individuals falling beyond plus or minus three standard deviations from the mean of the European sample cluster. PCs were recalculated for the filtered European test and replication cohorts (**S1 Fig** shows stratification of the European cluster along the first two principal components) and the first ten principal components fitted as covariates in all regression models.

### KIR imputation

The KIR$^*$IMP software [72] utilises 301 phased Immunochip SNP genotypes across the *KIR* locus (Chr19: 59,793,991–60,190,556, Hg18) to impute gene dosages across 14 *KIR* loci (*KIR2DP1, 2DS1, 2DS2, 2DS3, 2DS4, 2DS5, 2DL1, 2DL2, 2DL3, 2DL4, 2DL5, 3DP1, 3DL1, 3DS1)* against a UK reference panel of 479 *KIR* haplotypes. Framework genes *KIR3DL2* and *3DL3* are present in all common haplotypes and thus were not imputed or tested for disease associations in this study. *KIR3DL1* and *KIR3DS1* (*KIR3DL1/S1*), and *KIR2DL2* and *KIR2DL3*

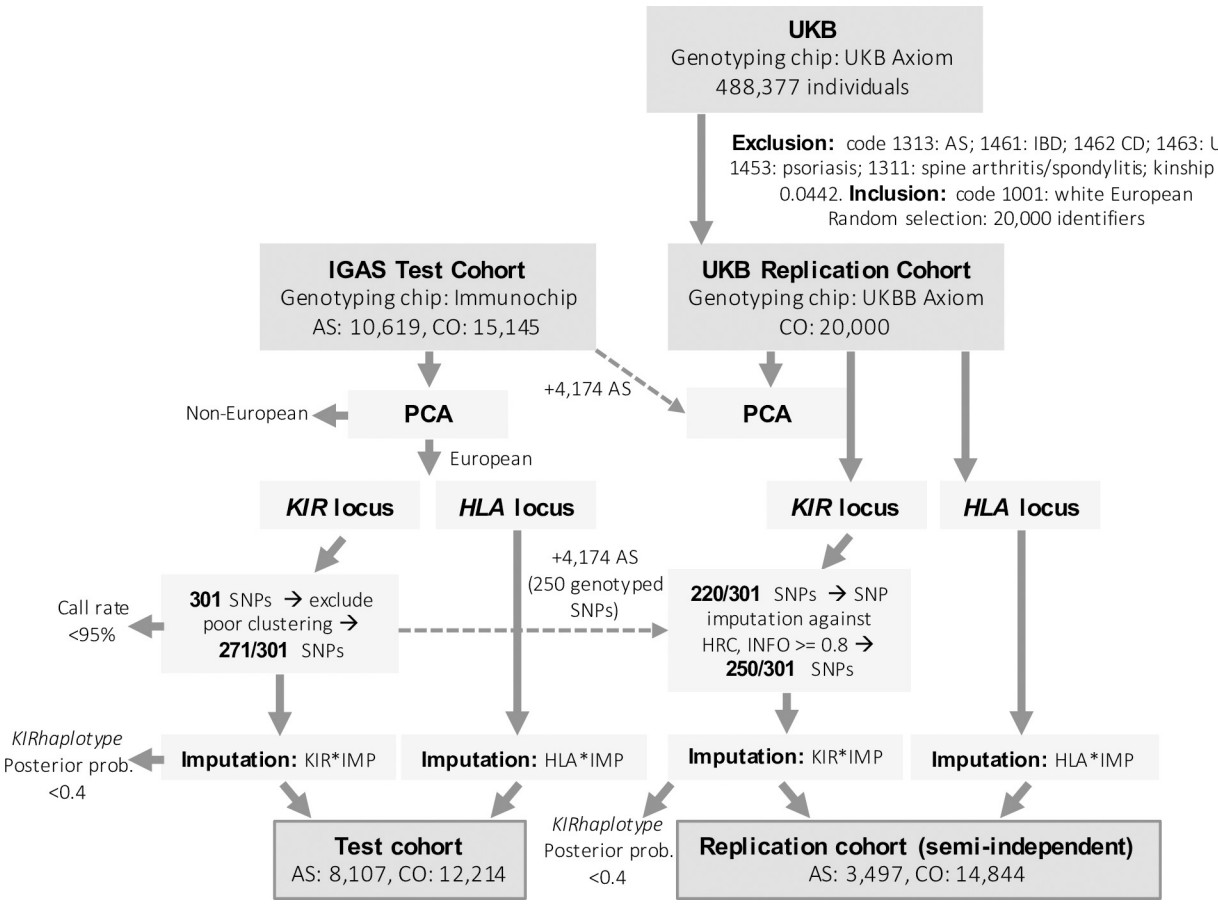

**Fig 1. Diagrammatic representation of study cohort inclusion and exclusion criteria.** 10,619 AS cases and 15,145 controls genotyped on the Illumina Immunochip and 20,000 UKB controls genotyped on the Affymetrix UKB Axiom array comprised the initial data set. KIR*IMP accepts genotypes across 301 *KIR* locus SNPs incorporated into the Immunochip array as a 'gold standard' SNP set; UKB Axiom Array data was SNP imputed in an attempt to increase SNP overlap with the reference set. A subset of 4,174 British IGAS AS cases from the test cohort were incorporated into the semi-independent replication cohort. To minimise bias attributed to *KIR* imputation on disparate SNP sets, these individuals had *KIR* dosages re-imputed using the reduced set of 250 SNPs available for UKB controls. Principal component analysis (PCA) was used to exclude ethnic outliers and calculate PCs to correct for population stratification in statistical tests. The final test cohort comprised 8,107 AS cases and 12,214, controls and the replication cohort 3,497 cases and 14,488 controls. AS: ankylosing spondylitis, IBD: inflammatory bowel disease, CD: Crohn's disease, UC: ulcerative colitis.

(*KIR2DL2/3*) are largely considered alleles of the same gene. Prior to imputation, for the IGAS test cohort only, genotype clustering across all 301 *KIR* locus SNPs was manually checked and adjusted (if necessary) using raw bead intensity files read into the Illumina GenomeStudio software. SNP genotypes that could not be unambiguously clustered were excluded, with 271/301 available SNPs remaining. Individuals with a call rate below 95% across the retained SNP set were removed. SNPs positions were converted to Hg19 build and genotypes phased using SHAPEIT [77]. Phased haplotype and sample files were passed to the online KIR*IMP server [72] for imputation, returning gene dosage calls (in the form of 0 or 1 count per chromosome, 2 in the case of gene duplication) and posterior probabilities for dosage calls. Raw bead intensity files were not available for reclustering of UKB replication cohort control samples and only 220 of the 301 *KIR* locus SNPs typed by the Immunochip were also accurately genotyped by the Affymetrix Axiom array. To increase SNP overlap with the KIR*IMP UK reference panel, the Sanger Imputation Server [78] was used to impute *KIR* locus genotypes for UKB controls against the Haplotype Reference Consortium (HRC) reference panel [78], increasing

available genotypes to 250/301 SNPs (INFO score > = 0.8) prior to KIR*IMP *KIR* dosage imputation as above. To reduce potential bias attributed to imputing *KIR* dosages with two differing SNP sets, the IGAS UK AS cases paired with the UKB controls in the semi-independent replication dataset were re-imputed on the reduced set of 250 *KIR* locus SNPs.

For both cohorts, individuals with an imputed '*KIRhaplotype*' posterior probability score <0.4 were excluded, largely removing those haplotypes with poor imputation of genes *KIR2DP1*, *KIR2DL1*, *KIR2DL5* or *KIR2DS3*, which were the least well typed from available genotype data by the KIR*IMP algorithm. Each phased gene content haplotype was annotated as according to Jiang *et al*. (2012) [23] as an 'A' or 'B' group haplotype (see **S1 Table** for definitions of gene content haplotypes), assuming presence of framework genes *KIR3DL3* and *KIR3DL2* in all haplotypes. Novel gene content haplotypes were specified with the prefix 'N'. *KIR3DL1* isoforms with variable splicing of exons 4 and 9 were distinguished by *KIR3DL1ex4* and *KIR3DL1ex9* imputed exon dosages respectively, however due to poor representation of rare *KIR3DL1* isoforms in the study cohorts only gene level *KIR3DL1* dosages were included in this analysis. The presence of a 22pb deletion in *KIR2DS4* was specified by the software as *KIR2DS4DEL*, with *KIR2DS4TOTAL* denoting the total dosage of the gene (i.e. *KIR2DS4DEL + KIR2DS4WT;* wildtype). KIR*IMP posterior probability scores for gene dosage imputation accuracy in the test and replication cohort are depicted in **S2 Fig**. A comparison of *KIR* gene frequencies and population prevalence taken from imputed test and replication cohorts, and frequencies reported for European populations in the Allele Frequency Net Database [79], is presented in **S3 Fig**. A comparison of *KIR* haplotype frequencies taken from imputed test and replication cohorts and those published in Jiang *et al*. (2012) [23] is presented in **S2 Table**.

## HLA imputation

HLA class I alleles were imputed at four-digit subtype resolution using HLA*IMP:03 [80]. Prior to allele imputation, genotypes were converted to Hg19 positions and Illumina plus strand orientation and the Michigan Imputation Server [81] was used to SNP impute *HLA* locus genotypes (Chr6: 20Mb-40Mb, Hg19) against the 1000 Genomes Phase 3v5 reference panel, with phasing using SHAPEIT [77]. The returned imputed SNP haplotype and sample files were passed to HLA*IMP:03 for *HLA* imputation, which outputs the presence or absence of each allele in each individual with a corresponding posterior probability of imputation accuracy. The *HLA* subclass of each allele (i.e. HLA-Bw4, Bw6, C1 or C2) was assigned based on known allele classifications (**Table 1**). The number of HLA alleles satisfying the following classes were summed per individual for use in statistical analyses: HLA-Bw4, HLA-Bw4A, HLA-Bw4B, HLA-Bw4B(I80), HLA-Bw4B(T80), HLA-C1 and HLA-C2.

### *KIR3DL1/S1* functional group typing

DNA previously extracted from whole blood or saliva was available for 236 *HLA-B*27+* AS patients and 99 *HLA-B*27+* control individuals from the test cohort. DNA concentrations were checked via Qubit fluorometric quantitation and normalised to 10ng/μL in UltraPure $H_2O$. Five separate PCR reactions designed to segregate *KIR3DL1* alleles into six distinct functional groups were performed on each sample as per the protocol published by Boudreau *et al*. [73]. In brief an array of primers were designed to target polymorphic sites within three exons of the *KIR3DL1* gene (exon 3, 4 and 7), combinations of which stratify alleles into two highly expressed (*High-1* and *High-2*), two lowly expressed (*Low-1* and *Low-2*), null and activating (*KIR3DS1*) groups (**S3 Table**). 50ng of input genomic DNA was amplified across each 25uL PCR reaction with reagent concentrations and cycling temperatures as described [73]. PCR products were run on a 1.5% agarose gel with 0.025μL/mL ethidium bromide at 125V for 40

minutes and visualised under UV light. Individuals were assigned a dosage between 0 and 2 for each of the six *KIR3DL1/S1* allele groups. Nine individuals who typed positive for three *KIR3DL1* alleles, likely due to locus duplication, were excluded from further analysis.

## Statistical analysis

All statistical analyses were conducted using custom scripts in R [82]. LD across the KIR locus was calculated using the 'LD' function from the package *genetics* [83] based on test cohort imputed haplotypes. A generalised linear model (GLM) with logarithmic link (logistic regression) was used to test for an association between imputed haplotype calls or KIR gene dosages and disease status, with inclusion of the first 10 principal components calculated per cohort for population stratification correction. When testing for an interaction between *KIR* and *HLA* subclasses or alleles an interaction term was included in the model. *KIR* dosage was assessed under a dominant (0 = gene absent, 1 = one or more gene copy) and recessive (0 = dosage less than two, 1 = homozygosity at locus) inheritance model. Given that some KIR genes harbour alternate alleles that differ in their activating or inhibitory classification, and genes are strongly linked in both positive and negative LD, dominant and recessive inheritance models were considered the most relevant when assessing the biological consequences of differing *KIR* dosages. *HLA* status was treated as dominant. All analyses were conducted in both test and replication cohorts, however replication cohort results are shown only for the top gene associations and epistatic interactions returned from the test cohort. Where applied, correction for multiple testing was performed using the Bonferroni method with statistical significance defined as analyses achieving P<0.05. For interaction P-values, multiple testing correction was applied across all tested KIR-HLA interactions, for KIR association P-values, multiple testing correction was applied across all tested KIRs within the specified HLA subtype or allele group. It is acknowledged that there is presently no established convention for a standard of evidence when reporting genetic interactions (whereas the genome-wide significance threshold is used for single-variant associations). We have thus used the conventional P-value threshold of P<0.05 following multiple testing correction (which controls the family-wise type 1 error rate) to highlight genetic epistatic interactions with greater evidence of disease association.

The association of *ERAP1* SNP rs30187 with disease risk was assessed under an additive model using a logistic regression in *KIR3DL1*+ or–and *KIR3DS1*+ or–cohorts, defined based on genotype at the top *KIR3DL1/S1* tag SNP rs592645 [72] (A allele = *KIR3DL1* present, T allele = *KIR3DL1* absent). An additional 26,607 *HLA-B*\*27+ and 299,267 *HLA-B*\*27- Caucasian controls from the UKB were added to the test cohort to boost sample size. Disease associations with *KIR3DL1* functional groups were assessed using a GLM as previously, under both dominant and recessive modes of inheritance.

## Results

### Study cohorts

Immunochip genotyping array data for 10,619 AS cases and 15,145 controls of mixed ethnicity from the IGAS Consortium, and UK Biobank (UKB) Axiom chip genotyping array data for 20,000 unrelated UKB controls of Caucasian ancestry without AS or associated inflammatory disease, were available prior to sample filtering and imputation quality control (**Fig 1**). We identified and excluded samples with cryptic relatedness, non-Caucasian ancestry, poor *KIR* locus SNP clustering or poor KIR\*IMP imputed *KIR* haplotype posterior probability scores. This left 8,107 AS cases and 12,214 healthy controls in the IGAS test cohort, and 3,497 IGAS AS cases and 14,844 UKB controls in the replication cohort. As the 3,497 AS cases in the replication cohort were selected from the British AS cases in the IGAS test cohort, to match the

ethnicity of the UKB controls, the replication dataset is considered only semi-independent. *KIR* haplotypes for these individuals were re-imputed using the reduced SNP set available for the UKB control samples (**Fig 1**). *KIR* gene and haplotype frequencies taken from imputed test and replication cohorts were largely in accordance with those reported for published European populations with dosages typed by laboratory-based methods. However, implementation of imputation quality control led to a slight bias in both cases and controls toward A group haplotypes, due to their lower copy-number variability and higher imputation accuracy relative to B group haplotypes (**S2 Table**, **S3 Fig**). A direct comparison of *KIR* dosages for 32 Centre d'Etude du Polymorphisme Humain (CEPH) and 20 control samples for which both imputed data and laboratory *KIR* typing was available revealed >90% concordance across loci with a per gene imputation posterior probability threshold of $>= 0.4$, and 100% at a threshold of $>= 0.6$ (**S4 Table**). All sample data, including disease status, imputed *KIR* gene and *HLA* allele dosages, *HLA* subtype counts and principal components are included in **S5 Table** and **S6 Table** for test and replication cohorts respectively.

## KIR gene associations with HLA-B*27+ AS

*KIR* gene associations with AS were assessed under a dominant and recessive mode of inheritance in the *HLA-B*27+ cases and controls. In the test cohort, there was some evidence of a disease-risk association with the dominant inheritance of *KIR2DS1*, *KIR2DL5*, *KIR3DS1* and *KIR2DS5* (P<0.05) prior to multiple testing correction (**Table 2**). *KIR2DS1*, *KIR3DS1* and *KIR2DS5* show strong positive linkage disequilibrium (LD; $R^2 > 0.7$, **S4 Fig**) and are inherited together with *KIR2DL5B* as part of the telomeric tB01 motif in a number of common B haplotypes (B3, B7 and B8, **S1 Table**). Under a recessive inheritance model, homozygosity for *KIR3DL1* and *KIR2DS4TOTAL* ($R^2 = 1$) was nominally associated with disease protection, and homozygosity for *KIR2DS5* with disease risk (**Table 2**). Haplotypes B3 and B11 were seen at increased frequency in HLA-B*27+ AS cases (**Table 2**). Multiple testing correction ablated KIR gene and haplotype associations in the test cohort and associations were not evident in the replication cohort.

**Table 2. *KIR* genes associations with *HLA-B*27+ AS.**

| | TEST COHORT | | | | REPLICATION COHORT | | | |
|---|---|---|---|---|---|---|---|---|
| Gene | AS Prop. (Count) | CO Prop. (Count) | OR | P | AS Prop. (Count) | CO Prop. (Count) | OR | P |
| 2DL5+ | 0.437 (3041/6952) | 0.400 (427/1067) | 1.16 | **0.03** | 0.381 (1134/2974) | 0.398 (487/1225) | 0.93 | 0.29 |
| 2DS1+ | 0.327 (2272/6952) | 0.293 (313/1067) | 1.17 | **0.03** | 0.326 (970/2974) | 0.330 (404/1225) | 0.98 | 0.73 |
| 2DS5+ | 0.256 (1780/6952) | 0.227 (242/1067) | 1.18 | **0.03** | 0.258 (766/2974) | 0.268 (328/1225) | 0.95 | 0.49 |
| 3DS1+ | 0.325 (2260/6952) | 0.292 (312/1067) | 1.17 | **0.03** | 0.324 (963/2974) | 0.328 (402/1225) | 0.97 | 0.70 |
| 2DS4T++ | 0.673 (4678/6952) | 0.707 (754/1067) | 0.85 | **0.03** | 0.674 (2004/2974) | 0.669 (820/1225) | 1.03 | 0.69 |
| 2DS5++ | 0.022 (153/6952) | 0.012 (13/1067) | 1.82 | **0.04** | 0.020 (58/2974) | 0.024 (30/1225) | 0.79 | 0.30 |
| 3DL1++ | 0.673 (4680/6952) | 0.707 (754/1067) | 0.85 | **0.03** | 0.674 (2004/2974) | 0.670 (821/1225) | 1.02 | 0.73 |
| Haplotype | AS Prop. (Count) | CTRL Prop. (Count) | OR | P | AS % (Prop.) | CTRL % (Prop.) | OR | P |
| B3[a] | 0.107 (1487/13904) | 0.093 (199/2134) | 1.17 | **0.04** | 0.109 (648/5948) | 0.120 (293/2450) | 0.89 | 0.13 |
| B11[b] | 0.011 (147/13904) | 0.005 (11/2134) | 2 | **0.03** | 0.010 (61/5948) | 0.009 (23/2450) | 1.05 | 0.86 |

[a] B3: *3DL3 – 2DL3 – 2DP1 – 2DL1 – 3DP1 – 2DL4 – 3DS1 – 2DL5A – 2DS5 – 2DS1 – 3DL2*

[b] B11: *3DL3 – 2DL3 – 2DP1 – 2DL1 – 3DP1 – 2DL4 – 3DS1 – 2DL5A – 2DS3 – 2DS1 – 3DL2*

+ = dominant inheritance

++ = recessive inheritance (homozygosity); Haplotype frequencies are calculated across all haplotypes (2n); AS = ankylosing spondylitis, CO = control,

*KIR2DS4T = KIR2DS4 TOTAL* (sum of wild-type and deletion alleles), OR = odds ratio, SE = standard error, P = P-value (bold = significant at P<0.05 prior to multiple testing correction), NS = not significant.

## KIR interactions with HLA subclass ligands

The association of co-inherited *KIR*s and *HLA* subclass ligands with AS risk was assessed using an interaction term in a generalised linear model. A significant *KIR-HLA* interaction term indicates that the association of a given *KIR* with disease risk is seen particularly in the context of a specific *HLA* subclass. *HLA* subclass ligand count was treated as dominant and *KIR*s assessed under both dominant and recessive modes of inheritance. Only interactions between *KIR*s and *HLA* subclass ligands known to biologically interact, and showing evidence of a disease association, are reported in **Table 3**, full results (showing *KIR*s in LD with reported genes and interactions between receptors and ligands without present functional evidence) are reported in **S7 Table**.

There was evidence for a statistical interaction between *KIR3DL1* and *HLA-Bw4A* (HLA-A alleles with the Bw4 motif) in both test and replication cohorts. Carriage of *KIR3DL1* in those who had inherited at least one *HLA-Bw4A* subclass allele was associated with increased disease risk, whereas no association was observed in *HLA-Bw4A* negative subjects (**Table 3**). Nominal interactions were also observed between *KIR3DL1* and *HLA-Bw4B(I80)*, whereby in the absence of an *HLA-Bw4B(I80)* ligand *KIR3DL1* homozygosity was associated with increased disease risk (**Table 3**). There was no evidence of an interaction between *KIR3DL1* and *HLA-Bw4B(T80)* group ligands, of which the most common AS-associated allele in European populations, *HLA-B*27:05*, is a member. There was also evidence of an interaction between the activating receptor gene *KIR2DS1* and alleles of the *HLA-C2* subclass, encoding canonical KIR2DS1 ligands. Carriage of *KIR2DS1* in the absence of an *HLA-C2* allele (i.e. *HLA-C1* homozygosity) was associated with reduced disease risk in the test cohort only (**Table 3**).

## KIR interactions with HLA class I alleles

Several *KIR-HLA* subclass interactions appeared to be driven by specific *HLA* class I alleles (**Table 4**). *KIR3DL1* exhibited a statistical interaction with the *HLA-Bw4A* allele *HLA-A*32*, with odds of disease increased >3 fold in those co-inheriting both the receptor and ligand.

**Table 3. Interactions between *KIR* genes and *HLA* class I subclasses.**

| KIR | HLA subclass | TEST COHORT | | | | | | | REPLICATION COHORT | | | | |
|---|---|---|---|---|---|---|---|---|---|---|---|---|---|
| | | AS Prop. (Count) | CO Prop. (Count) | OR | P | P (Adj) | Int.P | Int.P (Adj) | AS Prop. (Count) | CO Prop. (Count) | OR | P | Int.P |
| 3DL1+ | Bw4A+ | 0.975 (2302/2360) | 0.958 (3324/3471) | 1.77 | **0.0003*** | **0.002** | **0.0002*** | **0.009** | 0.979 (947/967) | 0.965 (3903/4044) | 1.71 | **0.03** | **0.02** |
| | Bw4A- | 0.961 (5525/5747) | 0.965 (8438/8743) | 0.91 | 0.27 | 0.51 | | | 0.962 (2433/2530) | 0.965 (10418/10800) | 0.92 | 0.46 | |
| 2DS1+ | C2+ | 0.330 (1790/5427) | 0.333 (2392/7193) | 0.98 | 0.66 | 0.99 | **0.04** | 0.26 | 0.322 (737/2291) | 0.333 (2861/8600) | 0.95 | 0.30 | 0.71 |
| | C2- | 0.319 (855/2680) | 0.351 (1763/5021) | 0.86 | **0.004*** | **0.02** | | | 0.323 (390/1206) | 0.329 (2052/6244) | 0.98 | 0.74 | |
| 3DL1++ | Bw4B(I80)+ | 0.654 (880/1345) | 0.667 (1879/2815) | 0.9 | 0.42 | 0.77 | **0.05** | 0.34 | 0.633 (305/482) | 0.670 (2145/3202) | 0.86 | 0.14 | **0.04** |
| | Bw4B(I80)- | 0.678 (4582/6762) | 0.658 (6186/9399) | 1.1 | **0.006*** | **0.02** | | | 0.685 (2065/3015) | 0.669 (7791/11642) | 1.07 | 0.11 | |

P = P-value denoting statistical significance of the KIR association with AS when assessed in the specified HLA subclass, Int.P = P-value for the KIR-HLA subclass interaction term, Adj = adjusted P-value

* = P-values that remain significant (P<0.05) after multiple testing correction

+ = dominant inheritance

++ = recessive inheritance (homozygosity), AS = ankylosing spondylitis, CTRL = control, OR = odds ratio, SE = standard error. Proportions are the proportion of individuals with the specified *KIR* genotype in cohorts split by HLA subclass carriage.

The association of *KIR3DL1* in *HLA-A\*32+* individuals remained evident following multiple testing correction in the test cohort and was independently detected in the replication cohort. The absence of *HLA-C\*05* (*HLA-C2*) specifically was associated with disease protection in *KIR2DS1* carriers. Alternatively, in the *HLA-C\*05+* cohort, carrying the deletion isoform of *KIR2DS4 (KIR2DS4D)* was also associated with disease protection, although evidence was lost with multiple testing correction. *KIR2DS1/4-HLA-C\*05* interactions were not evident in the replication cohort. *KIR2DS2/L2/L3* (*KIR2DS2* and *KIR2DL2* are in strong LD, $R^2 = 1$ **S4 Fig**, and *KIR2DL2* and *KIR2DL3* are alternate alleles of the same gene) showed evidence of an interaction with the *HLA-C1* alleles *HLA-C\*12, HLA-C\*07* and *HLA-C\*08* in the test cohort, though it was not possible to determine which of the tightly linked *KIR*s were driving the interaction effect. Interactions with *HLA-C\*12* were the strongest observed and of the nature that *HLA-C\*12+* individuals homozygous for *KIR2DS2* and *KIR2DL2* were at a two times increased risk of disease, whereas those carrying at least one copy of *KIR2DL3* were protected (**Table 4**). *KIR* associations tested in *HLA-C\*12+* individuals remained significant at P<0.05 upon multiple testing correction in the test cohort however trends were less evident in the smaller replication

**Table 4. Interactions between *KIR* genes and *HLA* class I alleles.**

| KIR | HLA | TEST COHORT | | | | | | REPLICATION COHORT | | | | |
|---|---|---|---|---|---|---|---|---|---|---|---|---|
| | | AS Prop. (Count) | CO Prop. (Count) | OR | P | P (Adj) | Int.P | AS Prop. (Count) | CO Prop. (Count) | OR | P | Int.P |
| 2DS2++ | HLA-C*12+ (C1) | 0.118 (52/441) | 0.061 (61/1000) | 2.05 | **0.0004*** | **0.003** | 0.0005 | 0.076 (11/144) | 0.063 (65/1029) | 1.22 | 0.56 | 0.58 |
| | HLA-C*12- (C1) | 0.070 (533/7666) | 0.070 (780/11214) | 0.99 | 0.90 | 0.96 | | 0.066 (221/3353) | 0.066 (916/13815) | 0.99 | 0.89 | |
| 2DL2++ | HLA-C*12+ (C1) | 0.118 (52/441) | 0.061 (61/1000) | 2.05 | **0.0004*** | **0.003** | 0.0006 | 0.076 (11/144) | 0.063 (65/1029) | 1.22 | 0.56 | 0.63 |
| | HLA-C*12- (C1) | 0.070 (535/7666) | 0.070 (780/11214) | 1.00 | 0.96 | 0.96 | | 0.067 (225/3353) | 0.066 (910/13815) | 1.02 | 0.85 | |
| 2DL3+ | HLA-C*12+ (C1) | 0.882 (389/441) | 0.939 (939/1000) | 0.49 | **0.0004*** | **0.006** | 0.0006 | 0.924 (133/144) | 0.937 (964/1029) | 0.82 | 0.56 | 0.62 |
| | HLA-C*12- (C1) | 0.930 (7131/7666) | 0.930 (10434/11214) | 1.00 | 0.96 | 0.96 | | 0.933 (3129/3353) | 0.934 (12904/13815) | 0.99 | 0.91 | |
| 2DS4D+ | HLA-C*12+ (C1) | 0.864 (381/441) | 0.812 (812/1000) | 1.47 | **0.02** | 0.24 | **0.02** | 0.833 (120/144) | 0.835 (859/1029) | 1.00 | 0.99 | 0.71 |
| | HLA-C*12- (C1) | 0.835 (6402/7666) | 0.833 (9338/11214) | 1.02 | 0.64 | 0.73 | | 0.828 (2777/3353) | 0.818 (11301/13815) | 1.07 | 0.17 | |
| 3DL1+ | HLA-A*32+ (Bw4A) | 0.986 (775/786) | 0.961 (832/866) | 3.10 | **0.001*** | **0.006** | 0.003 | 0.992 (360/363) | 0.965 (989/1025) | 4.28 | **0.02** | **0.02** |
| | HLA-A*32- (Bw4A) | 0.963 (7052/7321) | 0.963 (10930/11348) | 1.01 | 0.91 | 0.97 | | 0.964 (3020/3134) | 0.965 (13332/13819) | 0.97 | 0.76 | |
| 2DS1+ | HLA-C*05+ (C2) | 0.363 (358/985) | 0.330 (833/2525) | 1.16 | 0.06 | 0.37 | **0.003** | 0.363 (182/501) | 0.335 (1067/3181) | 1.13 | 0.22 | 0.09 |
| | HLA-C*05- (C2) | 0.321 (2287/7122) | 0.343 (3322/9689) | 0.90 | **0.002*** | **0.02** | | 0.315 (945/2996) | 0.330 (3846/11663) | 0.94 | 0.14 | |
| 2DS4D+ | HLA-C*05+ (C2) | 0.813 (801/985) | 0.843 (2129/2525) | 0.81 | **0.03** | 0.37 | **0.005** | 0.828 (415/501) | 0.817 (2598/3181) | 1.09 | 0.52 | 0.88 |
| | HLA-C*05- (C2) | 0.84 (5982/7122) | 0.828 (8021/9689) | 1.09 | **0.03** | 0.13 | | 0.828 (2482/2996) | 0.820 (9562/11663) | 1.06 | 0.28 | |
| 3DL1++ | HLA-B*27+ (Bw4BT80) | 0.673 (4680/6952) | 0.707 (754/1067) | 0.85 | **0.03** | 0.12 | **0.01** | 0.674 (2004/2974) | 0.670 (821/1225) | 1.02 | 0.73 | 0.33 |
| | HLA-B*27- (Bw4BT80) | 0.677 (782/1155) | 0.656 (7311/11147) | 1.10 | 0.14 | 0.36 | | 0.700 (366/523) | 0.669 (9115/13619) | 1.15 | 0.14 | |
| 2DL3++ | HLA-C*07+ (C1) | 0.569 (1578/2774) | 0.538 (3661/6810) | 1.14 | **0.003*** | **0.01** | 0.02 | 0.585 (767/1312) | 0.554 (4735/8553) | 1.14 | **0.03** | **0.05** |
| | HLA-C*07- (C1) | 0.535 (2851/5333) | 0.537 (2900/5404) | 0.99 | 0.79 | 0.84 | | 0.553 (1208/2185) | 0.559 (3514/6291) | 0.98 | 0.65 | |
| 2DL2+ | HLA-C*07+ (C1) | 0.432 (1197/2774) | 0.462 (3149/6810) | 0.88 | **0.004*** | **0.01** | 0.02 | 0.416 (546/1312) | 0.445 (3809/8553) | 0.89 | **0.04** | 0.07 |
| | HLA-C*07- (C1) | 0.466 (2484/5333) | 0.464 (2505/5404) | 1.01 | 0.78 | 0.93 | | 0.446 (975/2185) | 0.441 (2776/6291) | 1.02 | 0.69 | |

(*Continued*)

**Table 4.** (Continued)

| KIR | HLA | TEST COHORT | | | | | | REPLICATION COHORT | | | | |
|---|---|---|---|---|---|---|---|---|---|---|---|---|
| | | AS Prop. (Count) | CO Prop. (Count) | OR | P | P (Adj) | Int.P | AS Prop. (Count) | CO Prop. (Count) | OR | P | Int.P |
| 2DS2+ | HLA-C*07+ (C1) | 0.432 (1197/2774) | 0.462 (3149/6810) | 0.88 | **0.004*** | **0.01** | **0.02** | 0.415 (545/1312) | 0.447 (3819/8553) | 0.88 | **0.03** | **0.05** |
| | HLA-C*07- (C1) | 0.465 (2482/5333) | 0.464 (2505/5404) | 1.01 | 0.81 | 0.93 | | 0.448 (978/2185) | 0.441 (2777/6291) | 1.02 | 0.63 | |
| 2DS4D+ | HLA-C*01+ (C1) | 0.826 (2692/3258) | 0.848 (690/814) | 0.85 | 0.15 | 0.42 | **0.02** | 0.819 (1141/1393) | 0.827 (794/960) | 0.95 | 0.67 | 0.21 |
| | HLA-C*01- (C1) | 0.844 (4091/4849) | 0.830 (9460/11400) | 1.11 | **0.02** | 0.09 | | 0.835 (1756/2104) | 0.819 (11366/13884) | 1.12 | 0.08 | |
| 2DS4D++ | HLA-C*01+ (C1) | 0.362 (1178/3258) | 0.382 (311/814) | 0.92 | 0.32 | 0.46 | **0.04** | 0.341 (475/1393) | 0.369 (354/960) | 0.89 | 0.20 | **0.03** |
| | HLA-C*01- (C1) | 0.374 (1814/4849) | 0.352 (4012/11400) | 1.10 | **0.006*** | **0.05** | | 0.370 (779/2104) | 0.346 (4809/13884) | 1.11 | **0.03** | |
| 3DL1+ | HLA-A*24+ (BwA4) | 0.971 (1266/1304) | 0.955 (1849/1936) | 1.56 | **0.03** | 0.08 | **0.04** | 0.972 (494/508) | 0.967 (2096/2167) | 1.21 | 0.52 | 0.63 |
| | HLA-A*24- (Bw4A) | 0.964 (6561/6803) | 0.964 (9913/10278) | 1.01 | 0.94 | 0.98 | | 0.966 (2886/2989) | 0.964 (12225/12677) | 1.03 | 0.77 | |
| 2DS4W+ | HLA-A*11+ | 0.342 (360/1052) | 0.397 (577/1453) | 0.79 | **0.007** | 0.11 | **0.02** | 0.367 (169/460) | 0.421 (757/1797) | 0.79 | **0.03** | **0.05** |
| | HLA-A*11- | 0.390 (2752/7055) | 0.394 (4235/10761) | 0.99 | 0.69 | 0.74 | | 0.409 (1241/3037) | 0.409 (5341/13047) | 1.00 | 0.93 | |
| 3DL1++ | HLA-B*51+ (Bw4BI80) | 0.651 (322/495) | 0.691 (700/1013) | 0.82 | 0.10 | 0.39 | **0.03** | 0.625 (125/200) | 0.663 (751/1132) | 0.84 | 0.29 | 0.20 |
| | HLA-B*51- (Bw4BI80) | 0.675 (5140/7612) | 0.658 (7365/11201) | 1.09 | **0.007*** | **0.03** | | 0.681 (2245/3297) | 0.670 (9185/13712) | 1.05 | 0.23 | |
| 2DL3+ | HLA-C*08+ (C1) | 0.946 (385/407) | 0.920 (894/972) | 1.67 | **0.04** | 0.22 | **0.04** | 0.931 (161/173) | 0.940 (989/1052) | 0.81 | 0.53 | 0.68 |
| | HLA-C*08- (C1) | 0.927 (7135/7700) | 0.932 (10479/11242) | 0.92 | 0.16 | 0.49 | | 0.933 (3101/3324) | 0.934 (12879/13792) | 0.99 | 0.91 | |
| 2DS2++ | HLA-C*08+ (C1) | 0.054 (22/407) | 0.080 (78/972) | 0.60 | **0.04** | 0.47 | **0.04** | 0.069 (12/173) | 0.060 (63/1052) | 1.23 | 0.53 | 0.64 |
| | HLA-C*08- (C1) | 0.073 (563/7700) | 0.068 (763/11242) | 1.08 | 0.18 | 0.34 | | 0.066 (220/3324) | 0.067 (918/13792) | 0.99 | 0.88 | |

P = P-value denoting statistical significance of the KIR association with AS when assessed in the specified HLA allele group, Int.P = P-value for the KIR-HLA allele interaction term

* = P-values that remain significant (P<0.05) after multiple testing correction, Adj = adjusted P-value

+ = dominant inheritance

++ = recessive inheritance (homozygosity), AS = ankylosing spondylitis, CTRL = control, OR = odds ratio, SE = standard error, KIR2DS4D = KIR2DS4 deletion allele, KIR2DS4W = KIR2DS4 wild-type. Proportions are the proportion of individuals with the specified KIR genotype in cohorts split by HLA allele carriage.

cohort. There was a modest interaction observed between KIR3DL1 homozygosity and HLA-B*27 carriage in the test cohort alone, through statistical significance was lost with multiple testing correction. HLA-B*27+ individuals who were homozygous for KIR3DL1 showed protection from disease, whereas there was no evidence of a protective association in the HLA-B*27- cohort. Interactions between receptors and HLA alleles of the canonical ligand subclass are reported in Table 4, with full results (showing KIRs in LD with reported genes and interactions between receptors and ligands without present functional evidence) reported in S8 Table.

## KIR interactions with ERAP polymorphisms in HLA-B*27+ AS

Given that the HLA-B*27-bound peptide can disrupt the affinity of KIR3DL1 for HLA-B*27 ligands, the lead AS associated SNP in ERAP1 (rs30187), known to alter peptide trimming by

the aminopeptidase, was tested in a three-way interaction with *KIR3DL1/S1* in the *HLA-B*27+* test cohort. The rs30187 association with *HLA-B*27+* AS was assessed in *KIR3DL1+*, *KIR3DL1-*, *KIR3DS1+* and *KIR3DS1-* cohorts, tagging *KIR3DL1/S1* dosage using the SNP rs592645 to enable inclusion of an additional 26,607 *HLA-B*27+* and 299,267 *HLA-B*27-* European controls from the UKB to boost sample size. There was substantial evidence of the rs30187 association with *HLA-B*27+* AS irrespective of *KIR3DL1/S1* dosage (*KIR3DS1+* P = 1.0x10$^{-41}$, *KIR3DS1-* P = 1.3x10$^{-28}$, *KIR3DL1+* P = 2.0x10$^{-66}$ and *KIR3DL1-* P = 0.007; **Fig 2**), providing no support for a three-way interaction between AS-associated *ERAP1* genetic variation, *KIR3DL1/S1* and *HLA-B*27* influencing AS susceptibility. The reduction in the significance of the rs30187 association in the *HLA-B*27+ KIR3DL1-* cohort can likely be ascribed to the low number of AS cases in this group. As reported previously, there was no

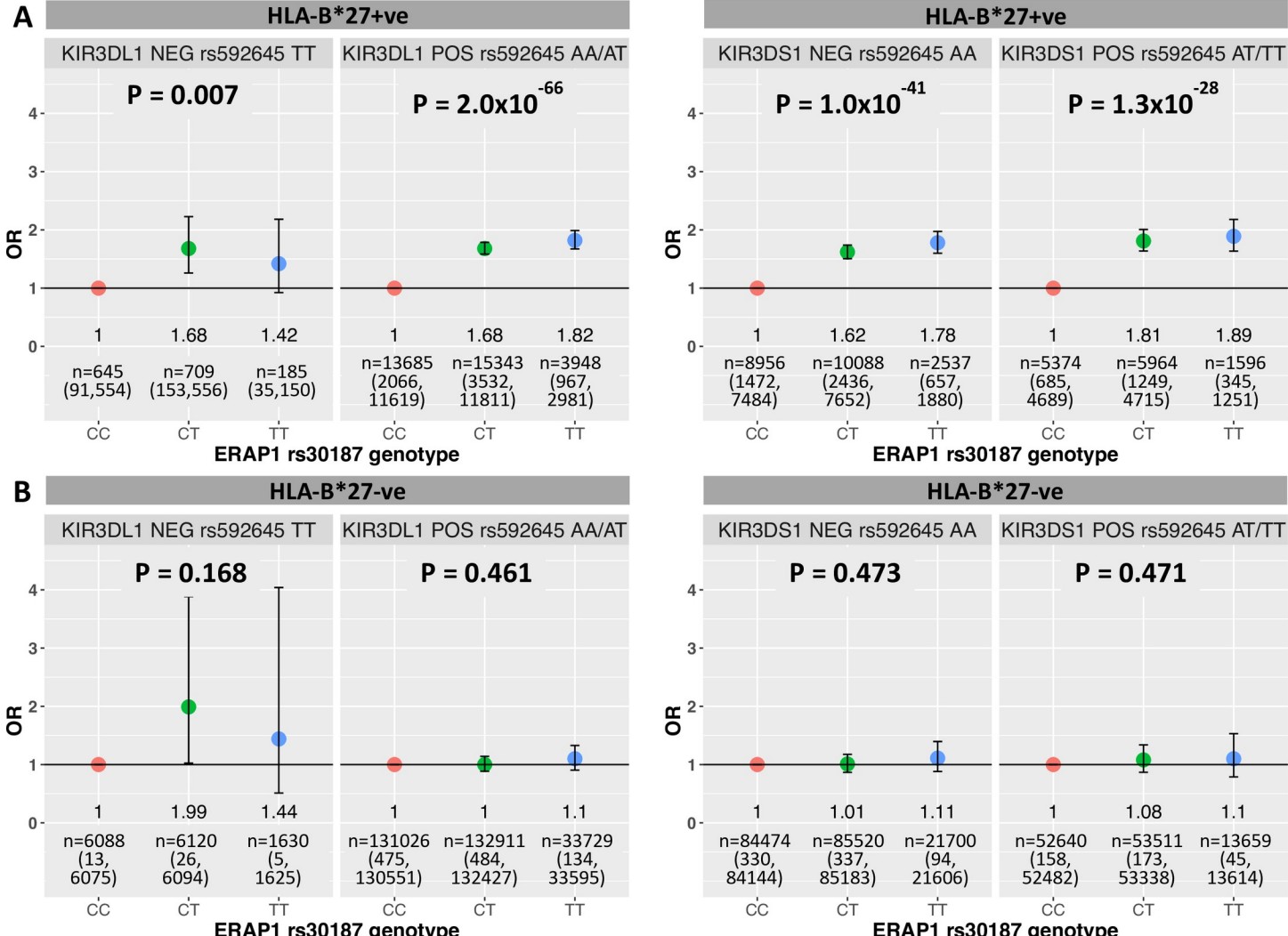

**Fig 2. Association of *ERAP1* SNP rs30187 with AS in *HLA-B*27+* and *HLA-B*27-* cohorts, split by carriage of *KIR3DL1* or *KIR3DS1*.** Points denote the OR of disease attributed to each successive increase in rs30187 risk allele [T] count in *HLA-B*27+* (**A**) and *HLA-B*27-* (**B**) individuals split by presence/absence of *KIR3DL1* (left) or *KIR3DS1* (right) defined using the *KIR3DL1/S1* locus tag SNP rs592645. P-values are derived from the logistic regression of genotype (additive) against disease status, ORs are derived from the exponentiated beta coefficient for each genotype class, treating the protective genotype rs30187 [CC] as baseline (OR = 1). The number of individuals in each *KIR-HLA-ERAP* genotype split group is indicated in the form n = total number (number of cases, number of controls). ORs are enumerated below points. Bars indicate 95% confidence interval.

**Table 5. Genotypic frequency of functional and non-functional *KIR3DL1* alleles and *KIR3DL1* functional groups in *HLA-B*27+ AS patient and controls.**

| Functional group | AS Prop. (Count) | CO Prop. (Count) | OR | P |
|---|---|---|---|---|
| 3DL1*f+ | 0.839 (198/236) | 0.869 (86/99) | 0.79 | 0.49 |
| 3DL1*f++ | 0.458 (108/236) | 0.414 (41/99) | 1.19 | 0.47 |
| 3DL1*n+ | 0.288 (68/236) | 0.303 (30/99) | 0.93 | 0.79 |
| 3DL1*n++ | 0.038 (9/236) | 0.020 (2/99) | 1.92 | 0.41 |
| Functional group | AS Prop. (Count) | CO Prop. (Count) | OR | P |
| 3DL1 High-1+ | 0.305 (72/236) | 0.384 (38/99) | 0.71 | 0.16 |
| 3DL1 High-1++ | 0.021 (5/236) | 0.051 (5/99) | 0.41 | 0.16 |
| 3DL1 High-2+ | 0.470 (111/236) | 0.424 (42/99) | 1.21 | 0.44 |
| 3DL1 High-2++ | 0.106 (25/236) | 0.071 (7/99) | 1.56 | 0.32 |
| 3DL1 Low-1+ | 0.305 (72/236) | 0.253 (25/99) | 1.30 | 0.33 |
| 3DL1 Low-1++ | 0.038 (9/236) | 0.030 (3/99) | 1.27 | 0.73 |
| 3DL1 Low-2+ | 0.051 (12/236) | 0.071 (7/99) | 0.70 | 0.48 |
| 3DL1 Low-2++ | 0.000 (0/236) | 0.000 (0/99) | NA | NA |
| 3DS1+ | 0.331 (78/236) | 0.354 (30/99) | 0.90 | 0.68 |
| 3DS1++ | 0.047 (11/236) | 0.040 (4/99) | 1.16 | 0.80 |

+ = dominant inheritance

++ = recessive inheritance (homozygosity), *f = functional, *n = null, AS = ankylosing spondylitis, CO = control, OR = odds ratio, P = P-value, NA = not applicable.

disease-association of rs30187 in *HLA-B*27-* individuals, irrespective of *KIR3DL1/S1* dosage (**Fig 2**).

## KIR3DL1/S1 allele group associations with HLA-B*27+ AS

The PCR protocol designed by Boudreau *et al.* (2014) was used to segregate *KIR3DL1* alleles from 236 *HLA-B*27+* AS patients and 99 *HLA-B*27+* controls into six distinct functional groups (Null, High-1, High-2, Low-1, Low-2 and *KIR3DS1*) based on a combination of polymorphic sites that segregate with surface expression phenotype [73]. High and low frequency alleles captured within each functional group are listed in **S3 Table**. Functional group frequencies in the control cohort were in accordance with those reported in the published method (**S3 Table**). When assessed under a dominant or recessive mode of inheritance, there was no significance difference in the frequency of functional cell-surface expressed (KIR3DL1*f: High-1, High-2, Low-1 and Low-2 alleles) or unexpressed (KIR3DL1*n: Null) *KIR3DL1* alleles between *HLA-B*27+ve* AS patients and controls. Similarly, there was no statistically significant difference in allele group frequency between patients and controls (**Table 5**). All data, including disease status and *KIR3DL1/S1* functional group typing is included in **S9 Table**.

## Discussion

Assessing the contribution of *KIR* and *HLA* co-inheritance to an immune-related phenotype is inherently challenging, with immense genomic diversity at both loci converging to influence the dynamics of an individual's lymphocyte responses. Very large cohorts are required to identify *KIR* associations with human disease amid the genetic 'noise' ascribed to copy-number variable haplotypes, and account for the modifying effect of independently inherited HLA ligands on KIR signalling. Here we have used imputation methods to type *HLA* class 1 alleles [80] and gene dosage level *KIR* haplotypes [72] from SNP genotype data in order to perform

the largest *KIR-HLA* association study reported to date. Encouragingly, paired comparison of 52 *KIR* typed and imputed Immunochip genotyped samples revealed >95% concordance at all but two loci (*KIR2DP1* and *KIR2DL1*). Prior to any quality filtering, the imputation of study cohorts returned comparable *KIR* haplotype and gene frequencies to directly typed European populations [23, 79]. However, removal of haplotypes with low (<0.4) posterior probability of imputation accuracy largely excluded individuals with poorly imputed rare gene content B group haplotypes. This led to a slight A group haplotype bias in both test and replication cohorts, hindering the ability to dissect genetic associations attributed to closely linked genes by conditional analysis (which are more likely to be found in rare arrangements on uncommon B group haplotypes). Albeit, given our sizeable dataset of high-confidence imputed haplotypes, we were able to detect suggestive interactions between *KIR* genes and *HLA* ligands at both subclass and allele resolution showing association with AS risk. It is possible that such interactions modify the activation threshold of lymphocyte populations in a disease setting, contributing to or sustaining a damaging autoinflammatory environment.

The profound association of the *HLA-Bw4* allele *HLA-B*27* with AS has encouraged investigation into a role for *KIR3DL1/S1* in disease; the inhibitory KIR3DL1 receptor being the only KIR known to engage canonical HLA-Bw4 ligands [40]. Its activating allele, KIR3DS1, has been demonstrated to mediate NK killing of HLA-B*57 (HLA-Bw4) target cells *in vitro* in a peptide dependent fashion [47], of potential additional interest in AS given the modest protective association of *HLA-B*57:01* with the disease [7], and may recognise other HLA-Bw4 ligands with similar peptide specificity. We identified a modest increase in the frequency of *KIR3DS1* in patients (29.2% in controls, 32.5% in AS) and genes *KIR2DL5*, *KIR2DS5* and *KIR2DS1* in the *HLA-B*27+* test cohort, all of which occur together on the telomeric half of haplotype B3 [23], which also showed a modest association with disease risk. Association of *KIR3DS1* with *HLA-B*27+* AS has been detected in numerous studies [57–59, 61, 64], with some studies reporting large *KIR3DS1* frequency differences between AS cases and healthy controls (e.g. 26.6% in controls and 64.3% in AS in a *HLA-B*27+* Chinese cohort of 72 individuals [58], and 40.4% in controls and 59.2% in AS in a *HLA-B*27+* Spanish cohort of 705 individuals [61]). Despite studying a much larger case-control dataset, we did not find a markedly increased frequency of *KIR3DS1* in AS patients in either the test or replication cohorts. Although differences in ethnic background may account for disparate results across studies, our data suggests that any association of *KIR3DS1* with AS must have quite a small overall effect size.

Conversely, homozygosity for *KIR3DL1* was nominally associated with disease protection in the *HLA-B*27+* test cohort, replicating previous studies demonstrating that the recessive genotype reduces disease risk [57, 58, 61, 64]. Despite being an HLA-Bw4T80 ligand with weak KIR3DL1 binding affinity relative to HLA-Bw4I80 alleles, the high surface density of HLA-B*27:05 (the predominant subtype in European HLA-B27 carriers) makes it a strong inhibitor of KIR3DL1+ NK cytotoxicity [27]. It could be hypothesised that the increased proportion of KIR3DL1+ NK cells in *KIR3DL1* homozygotes minimises the risk of undue lymphocyte activation and autoreactivity in healthy HLA-B*27 carriers. No *KIR3DL1* association was seen with *HLA-B*27-* disease, resulting in a detectable epistatic interaction between the two genetic factors in the test cohort, however the interaction was weaker than that seen between a number of other *KIR-HLA* pairs and significance was ablated with multiple testing correction.

It is possible that a protective effect of *KIR3DL1* in *HLA-B*27+* AS is confined to a specific allelic variant of the gene (for instance, engagement of HLA-B*27:05 inhibits NK cytotoxicity through KIR3DL1*002 but not KIR3DL1*007 [29], and a point mutation in KIR3DL1*004 disrupts protein folding and abolishes cell surface expression of this allotype altogether [84]), which would be obscured in gene-dosage resolution analyses. To assess this, we used a PCR

approach in a subset of the HLA-B*27+ test cohort to type and stratify *KIR3DL1* allotypes into six independent functional groups differing in surface expression and binding affinity [73]. We observed no disease-association with *KIR3DL1* genotype irrespective of whether functional (cell surface expressed) *KIR3DL1*f* or non-functional (*KIR3DL1*004)* *KIR3DL1* alleles were carried. Such finding differs from data presented by Zvyagin et al. (2010), showing disease protection afforded by carriage of the *KIR3DL1*f* allele and the *KIR3DL1*f* homozygous genotype, but no association with *KIR3DL1*004* in a HLA-B*27+ Caucasian cohort of 83 patients and 107 controls [64]. We also observed no significant disease-association with any specific *KIR3DL1* functional group, noting the power limitations given the modest sample size of this component of the study. This and previous studies have been underpowered to deconvolute *KIR3DL1* allelic associations at this locus [61], largely given the hyperpolymorphic nature of the receptor (for which 147 independent alleles are currently known [67]). Evidently, high-throughput KIR typing and allele imputation approaches need to advance in line with those available for the HLA locus if appropriately powered, high-resolution allotype association tests are to be feasible.

Additional to both the *KIR* allele and *HLA* subtype present [85], the outcome of KIR3DL1 signalling upon recognition of an HLA-B*27 ligand is known to be modified by the HLA bound peptide [34]. There is substantial genetic and functional evidence supporting a role for altered peptide processing in AS, particularly given a non-synonymous coding variant in *ERAP1*, rs30187, is uniquely associated with *HLA-B*27+* disease [4, 5] and modifies the HLA-B*27 peptidome [86, 87]. We tested the hypothesis that the rs30187 risk-association with *HLA-B*27+* disease is dependent on the *KIR3DL1/S1* background, with altered peptides either perturbing inhibitory signalling through KIR3DL1 or promoting NK activation through KIR3DS1 to enhance pathogenic lymphocyte responses. Disproving this, the rs30187 association with *HLA-B*27+* AS was clearly detectable in both *KIR3DL1-* and *KIR3DS1-* individuals. Although it cannot be disregarded that the altered HLA-B*27 peptidome shaped by disease-associated ERAP1 alleles may influence KIR3DL1/S1 signalling dynamics and subsequently lymphocyte activity, this does not appear to be the immunological mechanism by which the strong *ERAP1-HLA-B*27* epistasis in AS arises.

Dissecting genetic interactions between inherited *KIRs* and *HLA* class I alleles beyond HLA-B*27 requires a large enough cohort to retain a degree of statistical power when stratifying by both *KIR* and *HLA* carriage (some combinations of which are found at low frequency in the population). This study presents the first attempt to test for associations of *KIR-HLA* epistasis with an immune-mediated disease across all represented gene-gene combinations. Despite the substantial sample size of our test cohort, lack of an equally well powered replication cohort of independent AS patients made validation of findings difficult. Further, given strong LD across both the *HLA* and *KIR* locus, members of a statistically interacting receptor-ligand pair may not biologically interact but rather tag closely linked genes for proteins that do engage to alter immune cell function. Rare KIR haplotypes, in which unique gene combinations are observed, were likely overrepresented in those discarded during imputation quality filtering, hindering our ability dissect strong LD structures within this data. Here, only those interactions that can be interpreted in terms of experimentally validated KIR-HLA engagements are discussed, though to ascertain the relevance of these findings to AS pathogenesis on a functional level will require further experimentation. For the most part, this data supports a model whereby the contribution of many co-inherited *KIR* and *HLA* pairs shifts an individual's predisposition to AS, as is likely the case with many immune-mediated diseases, some combinations imparting more dominant effects than others.

In both test and replication cohorts, carriage of *KIR3DL1* in the presence of an *HLA-Bw4A* allele was associated with increased disease risk, the association remaining statistically

significant upon multiple testing correction in the former. At an allelic level, the *KIR3DL1* interaction could be detected most strongly with *HLA-A*32*, encoding a HLA-Bw4 ligand shown to protect target cells from lysis by KIR3DL1+ NK cells [88]. Why coinheritance of an inhibitory receptor-ligand pair may be associated with risk of, not protection from, a condition associated with chronic inflammation is unclear; HLA-A*32:01 binds with stronger affinity to some KIR3DL1 alleles than most HLA-Bw4B molecules [27, 28]. Increased inhibitory *KIR-HLA* binding strength has been shown to positively correlate with NK cytotoxicity against HLA-negative target cell lines *in vitro* [27], and this may exacerbate autoreactivity in instances that HLA expression is downregulated or altered peptides subvert binding in an inflammatory-disease setting. Numerous inflammatory diseases, including AS, have been associated with endoplasmic reticulum stress [89], which contributes to interrupted assembly and surface expression of appropriately folded HLA glycoproteins and increases target cell susceptibility to NK cell lysis [90].

In agreement with previous studies [57, 65], coinheritance of *KIR3DL1* and *HLA-B* ligands of the I80 subtype was decreased in AS patients, *KIR3DL1* homozygotes lacking an inhibitory *HLA-Bw4B(I80)* ligand significantly more often than controls in the test cohort. We did not detect a specific allele of the *HLA-Bw4B(I80)* subclass that appeared to drive this interaction. Strong KIR3DL1 binding to a number of HLA-Bw4B(I80) ligands [27, 28] may assist in reducing the activation threshold of NK and T-cell populations under homeostatic conditions in those who inherit both genetic factors. Absence of an HLA-C2 ligand in individuals who had inherited the activating receptor KIR2DS1 was also associated with disease protection in the test cohort. Unlike other activating receptors for which true ligands are unknown, KIR2DS1 recognition of HLA-C2 has been previously shown to activate KIR2DS1+ NK cells (with signalling sufficient to override KIR2DL1 inhibition on KIR2DL1+KIR2DS1+ cells), inducing IFNγ secretion and degranulation in a peptide dependent fashion [37, 91, 92]. Disease risk attributed to co-inheritance of KIR2DS1 and HLA-C2 has been previously observed in two independent AS cohorts [59, 65], as well as in psoriasis vulgaris [93, 94] and psoriatic arthritis [18], whereas it is associated with protection from Hodgkin's lymphoma [95] and the anti-leukemic activity of alloreactive NK cells in a HLA-C2 context [96]. Evidently this is a proinflammatory combination of factors. The *KIR2DS1-HLA-C2* interaction appeared to be driven most strongly by *HLA-C*05* in this study, perhaps suggesting that this ligand is a particularly potent activator of KIR2DS1+ NK or T-cells. Finally, we detected a number of *KIR2D/-HLA-C1* allele interactions associated with AS risk that are yet to be reported, however tight positive and negative LD between receptor genes *KIR2DS2*, *KIR2DL2* and *KIR2DL3* make it difficult to discern which receptor may be driving the biological effect. Strong disease-associations with activating and inhibitory *KIR2D* genes in *HLA-C*12* and *HLA-C*07* carriers in the test cohort, and suggestive associations in *HLA-C*01* and *HLA-C*08* carriers, emphasises that the HLA background likely imparts an array of modifying effects on disease risk beyond those ascribed to HLA-B*27 alone.

*In silico* prediction of *KIR* dosages from genotype data has opened avenues for *KIR* association testing in cohorts too large for conventional laboratory-based typing methods. Albeit, prediction algorithms inherently operate with some degree of error, and the findings of computational studies should be validated with laboratory methods where possible. Although *KIR* locus imputation is presently erroneous for rare gene content haplotypes, as large human reference datasets begin to capture the genomic diversity of the *KIR* locus these approaches will likely gain accuracy in both gene dosage and allelic discrimination. The targeted coverage of the *KIR* locus by the Illumina Immunochip has enabled us to impute high-confidence *KIR* dosages in a large test cohort of AS cases and controls for disease-association testing. We have endeavoured to replicate findings in a semi-independent cohort, incorporating UKB controls,

however the lower imputation accuracy (ascribed to reduced SNP coverage) and reduced size of this sample set has hindered this somewhat. Further replication of the results presented here are required to ensure accuracy and their biological interpretation will inevitably require functional investigation.

In conclusion, here we report the largest analysis of *KIR* and *KIR-HLA* co-associations with any immunological phenotype to date, addressing the contribution of these complex receptors to AS immunopathogenesis. We identified multiple nominally significant epistatic interactions between the genes encoding *KIR*s and their *HLA* ligands on both the subtype and allele level, suggesting that AS risk, as likely the case for many immune-mediated diseases, is modified by a mosaic of genetic effects that converge to influence the proinflammatory capacity of KIR expressing lymphocytes. Notably, although we replicated the direction of effect for previously reported *KIR3DL1/S1* associations with *HLA-B*27+ disease, we demonstrated that the *KIR3D* background does not modify the profound *HLA-B*27-ERAP1* epistasis observed in AS. Thus a three-way interaction between these factors is unlikely to be the primary driver of pathogenesis. As the resolution and throughput of KIR typing improves, more studies of this nature will assist in defining patterns in *KIR-HLA* coinheritance that contribute to complex immune phenotypes, improving understanding of the dynamic role of these receptors in health and disease.

## Supporting information

**S1 Table. Gene content of *KIR* haplotypes as defined by Jiang *et al*. (Genome Research. 2012;22:1845–1854.).** 'KIR Haplotype' corresponds to the *KIRhaplotype* number reported by *KIR*IMP* based on gene content, 'AvsB' provides the A/B haplotype classification. Dosage of each gene is indicated by colour (white = 0, pale green = 1, dark green = 2). *KIR2DS4D* = *KIR2DS4* deletion allele, *KIR2DS4W* = *KIR2DS4* wild-type, *KIR2DS4T* = *KIR2DS4 TOTAL* (sum of wild-type and deletion alleles). Isoforms of *KIR3DL1* with variable inclusion of exons 4 and 9 are indicated by *KIR3DL1ex4* and *KIR3Dl1ex9* respectively.
(DOCX)

**S2 Table. Frequency of top 19 *KIR* haplotypes in imputed test and replication cohorts compared to 2999 haplotypes derived from 793 families (Jiang *et al*. Genome Research. 2012;22:1845–1854.), with *KIR* dosage typed by quantitative PCR and haplotypes determined by segregation analysis in family groups.** Common haplotypes exceeding 1% frequency in the published dataset are shaded in grey.
(DOCX)

**S3 Table. Classification of *KIR3DL1* alleles into highly expressed, lowly expressed, null and activating *(KIR3DS1)* functional groups as defined by Boudreau *et al*. (PLoS One. 2014;9: e99543) with group frequencies in study cohort controls (2n = 198) and as published (2n = 852).** Proportions are calculated across all haplotypes from HLA-B*27+ controls in the study cohort (2n = 198). High and low frequency alleles are defined as in Boudreau *et al*.
(DOCX)

**S4 Table. Percent concordance between KIR*IMP imputed gene dosages and direct typing for a subset of control samples (including 32 CEPH samples).** Concordance was calculated on an individual rather than a haplotype basis as phasing was undetermined for laboratory typed samples. Percent concordance is reported either for all individuals (n = 52), or just those with all gene dosages imputed with posterior probability above 0.4 (n = 51) or 0.6 (n = 37). *KIR2DS4D* = *KIR2DS4* deletion allele, *KIR2DS4W* = *KIR2DS4* wild-type.
(DOCX)

**S5 Table. Test cohort sample data, including disease status, imputed *KIR* gene and *HLA* allele dosages, *HLA* subtype counts and principal components.**
(XLSX)

**S6 Table. Replication cohort sample data, including disease status, imputed *KIR* gene and *HLA* allele dosages, *HLA* subtype counts and principal components.**
(XLSX)

**S7 Table. Interactions between *KIR* genes and *HLA* class I subclasses (all interactions with interaction P-value = < 0.05).** Interactions in blue are those between receptors and HLA subclasses ligands known to biologically interact. Alternate shading indicates groups of KIRs in strong LD that demonstrate statistical interactions with the same HLA subclass ligand.
P = P-value denoting significance of the KIR association with AS when assessed in the specified HLA allele group, Int.P = P-value for the KIR-HLA allele interaction term, * = P-values that remain significant (P<0.05) after multiple testing correction, + = dominant inheritance, ++ = recessive inheritance (homozygosity), AS = ankylosing spondylitis, CTRL = control, OR = odds ratio, SE = standard error, NS = not significant, *KIR2DS4D = KIR2DS4* deletion allele, *KIR2DS4W = KIR2DS4* wild-type. Proportions are the proportion of individuals with the specified *KIR* genotype in cohorts split by HLA allele carriage.
(DOCX)

**S8 Table. Interactions between *KIR* genes and *HLA* class 1 alleles (all interactions with interaction P-value = < 0.05).** Interactions in blue are those between receptors and HLA ligands of a subclass known to biologically interact. Alternate shading indicates groups of KIRs (mostly in strong LD) that demonstrate statistical interactions with the same HLA allele.
P = P-value denoting significance of the KIR association with AS when assessed in the specified HLA allele group, Int.P = P-value for the KIR-HLA allele interaction term, * = P-values that remain significant (P<0.05) after multiple testing correction, + = dominant inheritance, ++ = recessive inheritance (homozygosity), AS = ankylosing spondylitis, CTRL = control, OR = odds ratio, SE = standard error, NS = not significant, *KIR2DS4D = KIR2DS4* deletion allele, *KIR2DS4W = KIR2DS4* wild-type. Proportions are the proportion of individuals with the specified *KIR* genotype in cohorts split by HLA allele carriage.
(DOCX)

**S9 Table. PCR typed KIR3DL1 functional groups from HLA-B\*27+ AS cases and controls.**
(XLSX)

**S1 Fig. PCA plot of European IGAS test cohort (AS patients and controls) and selected UKB replication cohort controls coloured according to dataset, IGAS (blue), UKB (orange).**
(TIF)

**S2 Fig. Average posterior probability of imputation accuracy across all imputed loci in the test and replication cohort.**
(TIF)

**S3 Fig. Comparison of *KIR* gene frequencies from *KIR\*IMP* imputed test and replication cohort controls and those averaged from a Serbian (n = 134) and Irish (n = 200) cohort reported in the Allele Frequency Net Database.** Bar charts show gene frequency comparisons for the test (A) and replication (B) cohorts with values enumerated in (C). *KIR2DL5*, *KIR3DS3* and *KIR2DS5* genes are duplicated on some B haplotypes and can occur on both the

centromeric and telomeric halves of the haplotype. At present *KIR*IMP* is unable to distinguish centromeric from telomeric copies of these genes so they have been grouped together as a single locus. The database did not include frequencies for wild type and deletion variants of *KIR2DS4*.
(TIF)

**S4 Fig. Pairwise LD between *KIR* genes calculated using KIR*IMP imputed haplotypes from test cohort controls.** Strength of LD is coloured according to $R^2$, with gene pairs in perfect positive or negative linkage (always or never occurring together in a haplotype) coloured red with an $R^2$ value of 1. Genes are ordered according to genomic position from *KIR2DS2* (centromeric) to *KIR2DS4* (telomeric), with exclusion of framework genes *KIR3DL3*, *KIR3DP1*, *KIR2DL4* and *KIR3DL2*. Distinction could not be made between centromeric and telomeric copies of *KIR2DS3/5* or *KIR2DL5*.
(TIF)

## Acknowledgments

This research was carried out at the Translational Research Institute (TRI), Woolloongabba, Queensland 4102, Australia. TRI is supported by a grant from the Australian Government. Members of the International Genetics of Ankylosing Spondylitis Consortium include: Paul Bowness, Paul Wordsworth: NIHR Oxford Musculoskeletal Biomedical Research Unit, Nuffield Orthopaedic Centre, Headington, Oxford, UK; Maxime Breban: INSERM UMR 1173, Université de Versailles Saint Quentin en Yvelines, Laboratoire d'excellence Inflamex, Saint-Quentn-En-Yvelines, France; Matthew Brown: School of Biomedical Sciences, Queensland University of Technology, Brisbane, Australia; Robert Colbert: National Institute of Arthritis and Musculoskeletal and Skin Diseases, NIH, Bethesda, Maryland, USA; Adrian Cortes: Nuffield Department of Clinical Neurosciences, University of Oxford, Oxford, UK; Bart Crusius: Department of Medical Microbiology and Infection Control, Laboratory of Immunogenetics, VU University Medical Center, Amsterdam, The Netherlands; Jing Cui, Soumya Raychaudhuri: Department of Medicine, Brigham and Women's Hospital, Boston, MA, USA; Dirk Elewaut: Department of Rheumatology, Gent University Hospital, Gent, Belgium; David Evans: University of Queensland Diamantina Institute, Princess Alexandra Hospital, Brisbane, Australia; Øystein Førre: Department of Rheumatology, Oslo University Hospital, and University of Oslo, Oslo, Norway; Dafna Gladman: Division of Rheumatology, University of Toronto, Toronto, Canada; Nigil Haroon, Robert Inman: Division of Rheumatology, Toronto Western Hospital, University of Toronto, Toronto, Canada; Benedicte Lie: Department of Medical Genetics, University of Oslo and Oslo University Hospital, Oslo, Norway; Carlos Lopez-Larrea: Department of Immunology, Hospital Universitario Central de Asturias, Oviedo, Spain; Walter Maksymowych: Department of Medicine, University of Alberta, Alberta, Canada; Javier Martin: Instituto de Parasitología y Biomedicina López-Neyra, Consejo Superior de Investigaciones-Científicas, Granada, Spain; Hans Nossent: School of Medicine, University of Western Australia, Perth, Australia Proton Rahman: Memorial University of Newfoundland, Newfoundland, Canada; John Reveille: Department of Rheumatology and Clinical Immunogenetics, University of Texas Health Science Center at Houston, Houston, Texas, USA; Fernando Santos: Chronic Diseases Research Centre (CEDOC), Faculdade de Ciências Médicas, Universidade Nova de Lisboa, Lisboa, Portugal; Simon Stebbings: Department of Medicine, Dunedin School of Medicine, University of Otago, Dunedin, New Zealand; Jaakko Tuomilehto: Department of Chronic Disease Prevention, National Institute for Health and Welfare, Helsinki, Finland; Rafael Valle-Oñate: Spondyloarthropaty Group-Division of Rheumatology, Hospital

Militar Central/Universidad de La Sabana, Bogotá, NA, Colombia; Irene van der Horst-Bruinsma: Department of Rheumatology, VU University Medical Centre, Amsterdam, Netherlands; Michael Weisman: Department of Medicine/Rheumatology, Cedars-Sinai Medical Center, Los Angeles, California, USA.

## Author Contributions

**Conceptualization:** Aimee L. Hanson, Tony J. Kenna, Matthew A. Brown.

**Data curation:** Aimee L. Hanson.

**Formal analysis:** Aimee L. Hanson.

**Funding acquisition:** Matthew A. Brown.

**Investigation:** Aimee L. Hanson, Jessica Harris, Tony J. Kenna, Matthew A. Brown.

**Methodology:** Aimee L. Hanson, Damjan Vukcevic, Stephen Leslie, Jessica Harris, Kim-Anh Lê Cao, Matthew A. Brown.

**Project administration:** Tony J. Kenna, Matthew A. Brown.

**Resources:** Tony J. Kenna, Matthew A. Brown.

**Supervision:** Kim-Anh Lê Cao, Tony J. Kenna, Matthew A. Brown.

**Visualization:** Aimee L. Hanson.

**Writing – original draft:** Aimee L. Hanson.

**Writing – review & editing:** Aimee L. Hanson, Damjan Vukcevic, Stephen Leslie, Jessica Harris, Kim-Anh Lê Cao, Tony J. Kenna, Matthew A. Brown.

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
