## [Decision Letter · Decision Letter 0]

10 Mar 2020

Dear Dr Brown,

Thank you very much for submitting your Research Article entitled 'Epistatic interactions between killer immunoglobulin-like receptors and human leukocyte antigen ligands are associated with ankylosing spondylitis' to PLOS Genetics. Your manuscript was fully evaluated at the editorial level and by independent peer reviewers. The reviewers appreciated the attention to an important problem, but raised some substantial concerns about the current manuscript. Based on the reviews, we will not be able to accept this version of the manuscript, but we would be willing to review again a much-revised version. We cannot, of course, promise publication at that time.

If you decide to revise the manuscript for further consideration at PLOS Genetics, please aim to resubmit within the next 60 days, unless it will take extra time to address the concerns of the reviewers, in which case we would appreciate an expected resubmission date by email to plosgenetics@plos.org.

[LINK]

We are sorry that we cannot be more positive about your manuscript at this stage. Please do not hesitate to contact us if you have any concerns or questions.

Yours sincerely,

Derry C. Roopenian

Associate Editor

PLOS Genetics

Scott Williams

Section Editor: Natural Variation

PLOS Genetics

Reviewer's Responses to Questions

**Comments to the Authors:**

Reviewer #1: In this research article, Hanson et al. investigated whether KIR/HLA interactions are associated with ankylosing spondylitis (AS) by using an imputation-based approach in a dataset of 12,214 healthy controls and 8,107 AS cases. This is a well-conducted study with a large dataset and the authors identified several nominal epistatic interactions between KIR gene and their ligands (at both HLA subtype and allele resolution) that increase risk of AS. However, most of these interactions failed to reach significance after multiple testing correction and were not replicated in a semi-independent cohort of 14,844 healthy controls and 3,497 cases (cases are a subset of the Discovery cohort). Thus it is difficult to draw definite conclusions regarding AS susceptibility.

Major comments:

1) My first concern relates to the identification of healthy controls in the UK biobank cohort. Depending on the age of the individuals in this cohort, it might be difficult to be sure that they won’t develop AS later, especially if they are B27 positive. It would have been interesting to include only patient of >40 years old.

2) Regarding KIR imputation: individuals with an imputed KIR haplotype posterior probability score <0.4 were excluded. The supplementary table 4 shows a perfect concordance between KIR imputed gene dosages and direct typing for a score >0.6. Why don’t you choose this threshold to increase the reliability of the imputation?

3) The multiple testing correction applied on the replication cohort should be explained in more details: have all the KIR/HLA interactions been tested only interactions with a significant pvalue in the Discovery cohort?

4) It would be interesting to give the power of this study to detect epistatic interactions

Minor comments:

1) The corrected p-values should be given in the table (and not only an asterisk indicating which p-values are <0.05 after correction).

Reviewer #2: This is the largest association analysis of KIR-HLA to investigate the contribution of these complex receptors to AS immune-pathogenesis. However, nominally significant epistatic interactions between the genes encoding KIRs and HLA ligands were identified. Also, they exclude the three-way interaction KIR3D background modify the HLA-B*27-ERAP1. The exact effect of KIR gene and pro-inflammatory capacity of KIR expressing lymphocytes remain elusive.

Comment:

1. Authors made a lot of KIR-HLA review that decipher the field of possible mechanisms. No significant confirm functional experiments were accomplished.

2. They identified several KIR2D/HLA-C1/C2 allele interactions beyond HLA-B27. As authors state a statistically interacting receptor-ligand pair may not biologically interact but rather tag closely linked genes. The effects of combinations interaction interpretation must be carefully.

3. What are the imputation weakness of HLA and KIR typing to gain accuracy since replication cohort show no significant findings?

4. What are the ethnic background difference?

**Have all data underlying the figures and results presented in the manuscript been provided?**

Reviewer #1: Yes

Reviewer #2: No:

PLOS authors have the option to publish the peer review history of their article (what does this mean?). If published, this will include your full peer review and any attached files.

Reviewer #1: No

Reviewer #2: No

---

## [Editor Report · Decision Letter 1]

3 Jun 2020

Dear Dr Brown,

We are pleased to inform you that your manuscript entitled "Epistatic interactions between killer immunoglobulin-like receptors and human leukocyte antigen ligands are associated with ankylosing spondylitis" has been editorially accepted for publication in PLOS Genetics. Congratulations!

Yours sincerely,

Derry C. Roopenian

Associate Editor

PLOS Genetics

Scott Williams

Section Editor: Natural Variation

PLOS Genetics

Comments from the reviewers (if applicable):

**Data Deposition**

http://datadryad.org/submit?journalID=pgenetics&manu=PGENETICS-D-20-00016R1

**Press Queries**

---

## [Editor Report · Acceptance letter]

22 Jul 2020

PGENETICS-D-20-00016R1 

Epistatic interactions between killer immunoglobulin-like receptors and human leukocyte antigen ligands are associated with ankylosing spondylitis 

Dear Dr Brown, 

We are pleased to inform you that your manuscript entitled "Epistatic interactions between killer immunoglobulin-like receptors and human leukocyte antigen ligands are associated with ankylosing spondylitis" has been formally accepted for publication in PLOS Genetics! Your manuscript is now with our production department and you will be notified of the publication date in due course.

With kind regards,

Kaitlin Butler

PLOS Genetics

On behalf of:
